# Healthcare Experience of Pediatric Patients with Autism Spectrum Disorders in Saudi Arabia: A Cross-Sectional Study

Basma Al-Jabri [1] , Sara Alnuwaiser [2,*] , Haifa Abdulghaffar [2], Rahaf Almuhanna [2] , Shaimaa Salaam [2], Raval Brika [2], Alia Addas [2] and Hala Bedaiwi [2]

1 Pediatric Department, Faculty of Medicine, King Abdulaziz University, King Abdulaziz University Hospital, Jeddah 21589, Saudi Arabia; baljabri@kau.edu.sa
2 Faculty of Medicine, King Abdulaziz University, Jeddah 21589, Saudi Arabia
* Correspondence: s.alnuwaiser@gmail.com

**Abstract:** Children with autism spectrum disorder (ASD) face several challenges in the healthcare setting. This study defines the challenges experienced by children with autism in hospitals in Saudi Arabia. A cross-sectional study was conducted using a questionnaire for guardians of autistic children in outpatient clinics, autism support groups, and rehabilitation centers. A total of 199 participants were included. The medical procedures causing the most anxiety to children were injections and getting their blood drawn (68.3%), vital sign measurement (41.6%), and height and weight measurement (37.8%). Long waiting hours (44.1%), increased sensory stimuli (33.2%), and overcrowding of hospital staff (27.9%) were stress-inducing in the healthcare environment. The guardians recommended that loud noises (44.7%), crowdedness (41.2%), and long waiting hours (42.1%) be avoided. The nonverbal children experienced significantly higher levels ($p < 0.001$) of agitation, irritability, and outbursts during doctor visits than their verbal counterparts. The children with intellectual disabilities were more tense and unresponsive during doctor visits (33.3%) than their intellectually able counterparts, who more frequently were calm and responsive (44.9%) during visits. Most patients with ASD face hardships during hospital visits. Nonverbal patients and those with intellectual disabilities have a higher tendency for hospital setting anxiety-induced outbursts, which may be eased by avoiding loud noise and overcrowding.

**Keywords:** autism spectrum disorder; healthcare setting; nonverbal; intellectual disability

## 1. Introduction

Autism spectrum disorder (ASD) is a neurodevelopmental disorder that significantly affects communication and social interactions and is associated with patterns of repetitive, restricted, or sensory behaviors [1]. In addition, children with autism are more prone to developing medical conditions such as epilepsy and gastrointestinal disorders [2]. They are almost four times more likely to visit the emergency department (ED) and have a higher risk of hospitalization than their peers [3]. Overall, children with autism have higher rates of visits to medical facilities and utilization of healthcare services [4,5].

Despite their frequent encounters with healthcare settings, children with autism face several challenges during hospital visits [6,7]. Children with ASD can often become anxious, defensive, and hyperstimulated in a healthcare setting, leading to tantrums and aggressive behaviors toward themselves and healthcare providers and making visits more challenging [8]. In particular, children with autism that have intellectual disabilities and hypersensitivity variants are more likely to have fearful experiences in hospital settings [9]. Despite these documented difficulties, healthcare systems are often not adequately equipped to support their unique sensory, behavioral, and communication needs [10]. A qualitative study by pediatric registered nurses in Santa Monica, Los Angeles identified a knowledge deficit regarding ASD (including a lack of knowledge of the different presentations and

classifications of the disorder) and not knowing the ASD patient as two major hindrances to providing care [11]. The participants in this study largely attributed this deficit to inadequate education during nursing school and a lack of continuing education services for ASD [11]. A cross-sectional study in Russia revealed that experienced healthcare workers lacked proper communication and examination skills for children with behavioral disorders. In addition, 83.5% of caregivers of autistic children believed that better individualized services were required during follow-up visits [12].

The estimated incidence of ASD in the Kingdom of Saudi Arabia is 2.81 per 1000 children [13]. A cross-sectional study in the Kingdom documented that parents believe there is a lack of appropriate social support and public understanding of their children's needs [14]. While there is an abundance of research describing children with autism in healthcare settings worldwide, little is known about the experiences of children with autism in healthcare settings in Saudi Arabia. The high and increasing prevalence of ASD among Saudi children calls for strategic planning and adequate preparation to address the healthcare needs of this at-risk population [13]. Additionally, tertiary hospitals in the Kingdom are not structured as "child-friendly" hospitals, nor are they optimized for those with neurodevelopmental disabilities such as autism [15]. Therefore, we aimed to describe the overall experiences of children with ASD in hospital-based healthcare settings in Saudi Arabia and identify the possible challenges facing them, their caregivers, and healthcare providers in an effort to facilitate the delivery of high-quality, evidence-based medical care.

## 2. Materials and Methods

We conducted a cross-sectional study using electronic questionnaires completed by the participants directly or over the phone by the researchers to explore the experience of children with autism in hospital-based healthcare settings and identify their challenges when seeking medical care. The standard method of data collection was through phone calls, however some participants elected to answer individually at their convenience. It was emphasized to each participant in this case that the researchers were available to answer any questions requiring further clarification. The study protocol was approved by the Research Ethics Committee of King Abdulaziz University (Approval No.: 157-21).

The participants in this study were caregivers, including parents and legal guardians of children with autism attending outpatient clinics, autism support groups, and rehabilitation centers for children with autism in Jeddah, Saudi Arabia. Our study population was selected through cluster sampling following a thorough search of the hospital's database for patients between the ages of 1 and 18 years diagnosed with ASD. To be included in the study, a participant had to be a primary caregiver of an autistic child or adolescent (18 years or younger) with a formal diagnosis of ASD made by a specialized physician.

We used an electronic questionnaire that included yes-or-no, multiple-choice, and open-ended questions. The questionnaire focused on the child's demographics, autism diagnosis, skills, health status including comorbidities, and behavior in different healthcare settings (inpatient, outpatient, and emergency department). A literature review was conducted to develop questions suited to the research hypothesis. A questionnaire previously validated and published by Muskat et al. (2015) [6] was used as a guide for creating this questionnaire. This was reviewed by two developmental pediatrics consultants for content validity, and 10 local parents were tested for face validity. Both Arabic and English versions were available based on the participant's preference. Informed consent was obtained from each participant prior to filling out the questionnaire.

The children were divided according to gender, age, and verbal and intellectual ability. Children were considered nonverbal if they preferred nonverbal communication or had poor verbal communication skills. Intellectual disability is a deficit in intellectual functioning (measured as an intelligence quotient (IQ) of less than 70) [16] as well as a deficit in adaptive functioning, wherein there is an inability to perform age-appropriate daily life activities that are present from childhood [15]. Thus, for the children with unknown IQ scores, we defined intellectual disability as any child over the age of 4 years [17] who

was never able to independently perform tasks of daily living and ranked as having "below-average intelligence" by his or her caregiver.

We summarized the clinical and demographic characteristics of the patients using descriptive statistics. Continuous variables are presented as the mean and standard deviation (Mean $\pm$ SD). Categorical variables are presented as total numbers and percentages. The chi-square test ($\chi^2$) was used to examine the relationship between the children's demographics and anxiety. Content analysis was performed for the open-ended questions. The nonparametric variables were tested using the Kruskal–Wallis test. Statistical significance was set at $p < 0.05$. The Statistical Package for the Social Sciences version 26 was used for the analysis [18].

## 3. Results

### 3.1. Respondent Characteristics

In total, 199 responses were obtained. The majority of the respondents (71.4%) were mothers of children with autism.

### 3.2. Children's Characteristics

Of the 199 children, 150 (75.4%) were male and 49 (24.6%) were female. The age ranges were 2–3 years (20 toddlers; 10.1%), 4–6 years (49 preschool-aged; 24.6%), 7–11 years (76 school-aged; 38.2%), and 12–18 years (54 adolescents; 27.1%). The average age at diagnosis was 3.22 $\pm$ 1.46 years.

Eight (4%) children had physical limitations, disabilities, or both. Furthermore, the majority ($n = 187$) of the children had comorbidities that could require frequent healthcare visits (Table 1), most notably attention deficit hyperactivity disorder (ADHD) (80.9%). A total of 61 children were reported to have sensory processing sensitivity, the most frequently reported one being touch and sound (equally 50.8%) followed by lighting (40.98%) and smell (3.28%).

**Table 1.** Comorbidities.

| Variable | No. (%) |
|---|:---:|
| Seizures or epilepsy | 16 (8) |
| Attention deficit hyperactivity disorder (ADHD) | 161 (80.9) |
| Constipation or selective eating disorder (SED) | 108 (54.3) |
| Sleep disturbances | 73 (36.7) |
| Agitation or nervousness | 84 (42.4) |
| Chest allergy or asthma | 27 (13.6) |
| He or she does not suffer from any other disorders | 11 (5.5) |

In terms of the children's IQ, 66.3% ($n = 132$) had taken an IQ test with a mean of 57.56 $\pm$ 17.27. In terms of the children with no prior testing, 53.7% ($n = 36$) of their parents believed that they had a "below-average IQ". Of the 160 children aged 5 years and older, most ($n = 61$; 38.1%) were able to perform their daily tasks.

When determining their ability to perform a daily routine, 39 participants were excluded because they were 4 years old or younger. Of the 160 left, most (38.1%; $n = 61$) were able to perform daily tasks but sometimes needed assistance, while 30.6% ($n = 49$) could not complete activities alone and required help at all times.

The majority of the children (63.3%) sometimes enjoyed talking to others. However, nonverbal communication was the preferred means of communication for 45.2% of these children. Most parents rated their children's abilities in both verbal and nonverbal communication as weak (25.1%) and very good (48.7%), respectively.

A total of 168 (84.4%) participants agreed that their children exhibited behaviors that made them anxious, presenting most frequently with anger, aggression, or violence toward themselves or others (Table 2).

**Table 2.** Anxiety and coping skills.

| | No. (%) |
|---|---|
| *Effects of Anxiety on the Child:* | |
| Anger, aggression, or violence toward oneself or others | 127 (63.8) |
| Repetitive behaviors | 84 (42.2) |
| Social withdrawal | 81 (40.7) |
| Selective mutism | 29 (14.6) |
| Rigidity of thinking | 28 (14.1) |
| *Does the child have skills to cope with anxiety?* | |
| No | 143 (71.9) |
| Yes | 56 (28.1) |
| *Coping skills:* | |
| Playing with toys or electronics | 10 (5) |
| Watching TV or using electronics | 7 (3.51) |
| Sitting in isolation | 5 (2.5) |
| Moving around | 5 (2.5) |

*3.3. Characteristics of the Healthcare Experiences*

Out of the total responses, 185 (93.0%) participants reported having healthcare experiences in Jeddah, while 14 (7.0%) reported having their experiences in other areas in the Kingdom. Outpatient clinic visits accounted for 59.8% (*n* = 119) of the children's healthcare experiences, of which 29.6% (*n* = 59) occurred within the past month and 30.2% (*n* = 60) occurred within the past year. On average, the children waited 1–2 hours (47.3%, *n* = 56) before being attended to. Approximately 27.7% (*n* = 55) of the children had been to the emergency department in the last year, and the average waiting time ranged from less than an hour (27.2%, *n* = 15) to 1–2 h (27.2%, *n* = 15). About 15.6% (*n* = 31) of the children were hospitalized at least once for less than one week in the past year, of which the majority (80.8%, *n* = 25) remained inpatients for less than one week. Many of them were responsive in the healthcare setting, with 34.7% of the children showing calm behavior in a hospital setting. However, 42.5% of the 74.4% (*n* = 148) who underwent phlebotomies experienced agitation, irritability, or outbursts. The data obtained showed that the medical procedure which caused the most anxiety in the children was the phlebotomy (68.3%), vital measurements (41.6%), and height and weight measurements (37.8%). In addition, when asked about the factors that made their children anxious in a medical setting, long waiting hours (44.1%), increased sensory factors in the hospital environment (33.2%), and overcrowding of hospital employees around the child (27.9%) were the top three factors. Most guardians recommended that loud noises (44.7%), crowdedness (41.2%), and long waiting hours (42.1%) should be avoided during visits to the hospital, clinic, or emergency department. Many participants subjectively rated the nursing and medical staffs' ability to manage children with autism as excellent (31.7% and 39.2%, respectively), whereas others rated the nursing and medical staff as weak (16.1% and 12.6%, respectively).

There was no statistically significant difference between genders in the reaction to doctor visits or phlebotomies (*p* = 0.113 and *p* = 0.618, respectively). The children of school-age experienced more agitation, irritability, and outbursts (42.9%) during phlebotomies than any other age group (*p* = 0.001), a reaction that was significantly lower than that of the adolescent group (17.5%). The results of anxious reactions across age groups are summarized in Table 3.

**Table 3.** Difference in anxious reactions across age groups.

| Variable | Age Group | | | | $\chi^2$ | *p* Value |
|---|---|---|---|---|---|---|
| | Toddlers (2–3 y) | Pre-School (4–6 y) | School Age (7–11 y) | Adolescent (12–18 y) | | |
| *How do the effects of anxiety appear in the child? (You may choose more than one answer):* | | | | | | |
| Repetitive behaviours | 8 (9.5) | 21 (25) | 32 (38.1) | 23 (27.4) | 0.05 | 0.997 |
| Social withdrawal | 11(13.6) | 19 (23.5) | 29 (35.8) | 22 (27.2) | 1.97 | 0.578 |
| Anger, aggression, or violence toward oneself or others | 10 (7.9) | 33 (26) | 49 (38.6) | 35 (27.6) | 1.95 | 0.582 |
| Rigidity of thinking | 2 (7.1) | 6 (21.4) | 13 (46.4) | 7 (25) | 1.04 | 0.791 |
| Selective mutism | 1 (3.4) | 7 (24.1) | 13 (44.8) | 8 (27.6) | 1.87 | 0.6 |
| *How did the child react previously when visiting the doctor?* | | | | | | |
| Tense and unresponsive | 7 (15.5) | 10 (25) | 13 (32.5) | 10 (25) | | |
| Tense and responsive | 5 (10.2) | 11 (22.4) | 20 (40.8) | 13 (26.5) | | |
| Very agitated and irritable, had an outburst | 5 (16.1) | 9 (29) | 13 (41.9) | 4 (12.9) | 19.65 | 0.074 |
| Calm and responsive | 2 (2.9) | 14 (20.3) | 26 (37.7) | 27 (39.1) | | |
| Calm and unresponsive | 1 (10) | 5 (50) | 4 (40) | 0 (0.0) | | |
| *Did the child have his or her blood drawn previously?* | | | | | | |
| No | 11 (21.6) | 10 (19.6) | 21 (41.2) | 9 (17.6) | 12.18 | 0.007 |
| Yes | 9 (6.1) | 39 (26.4) | 55 (37.2) | 45 (30.4) | | |
| *If the answer to the previous question was yes, how did he or she react?* | | | | | | |
| Tense and unresponsive | 0 (0.0) | 5 (21.7) | 7 (30.4) | 11 (47.8) | | |
| Tense and responsive | 1 (3.2) | 8 (25.8) | 10 (32.3) | 12 (38.7) | | |
| Very agitated and irritable, had an outburst | 5 (7.9) | 20 (31.7) | 27 (42.9) | 11 (17.5) | 36.96 | 0.001 |
| Calm and responsive | 1 (4.2) | 2 (8.3) | 11 (45.8) | 10 (41.7) | | |
| Calm and unresponsive | 2 (28.6) | 4 (57.1) | 0 (0.0) | 1 (14.3) | | |

The children were subdivided into verbal (*n* = 83), nonverbal (*n* = 116), intellectually able (*n* = 81), and intellectually disabled (*n* = 118) groups. There was no significant difference in communication between the genders, with 42.8% of females and 63.3% of males (*p* = 0.162) classified as nonverbal. When looking at the behavioral presentation of anxiety, the children who were nonverbal were more likely (*p* < 0.05) to present anger, aggression, or violence toward themselves or others (Table 4). The nonverbal children experienced significantly higher levels (*p* < 0.001) of agitation, irritability, and outbursts during doctor visits than their verbal counterparts, but there was no statistically significant difference in their reaction to phlebotomies. There was no statistically significant difference in the presentation of anxiety between the intellectually abled and disabled groups (Table 5). A greater number of children with intellectual disabilities were tense and unresponsive during doctor visits (33.3%) in comparison with their intellectually able counterparts, who more frequently were calm and responsive (44.9%) during visits. The difference in the reactions between the two groups was statistically significant (*p* < 0.001). As for reactions to blood draws, both groups were mostly agitated and irritable, with the greatest difference being that a calm and responsive reaction was more prevalent in the children without intellectual disabilities (22.0%) versus those with intellectual disabilities (10.2%). There was a statistically significant difference (*p* < 0.05) in the reactions between groups.

**Table 4.** Difference in anxious reactions between verbal and nonverbal communicators.

| Variable | Communication | | $\chi^2$ | *p* Value |
| --- | --- | --- | --- | --- |
| | Verbal | Nonverbal | | |
| *How do the effects of anxiety appear in the child? (You may choose more than one answer):* | | | | |
| Repetitive behaviors | 31 (36.9) | 53 (63.1) | 1.38 | 0.24 |
| Social withdrawal | 34 (42) | 47 (58) | 0.004 | 0.95 |
| Anger, aggression, or violence toward oneself or others | 60 (47.2) | 67 (52.8) | 4.42 | 0.035 |
| Rigidity of thinking | 10 (35.7) | 18 (64.3) | 0.48 | 0.488 |
| Selective mutism | 12 (41.4) | 17 (58.6) | 0.002 | 0.969 |
| *How did the child react previously when visiting the doctor?* | | | | |
| Tense and unresponsive | 7 (17.5) | 33 (82.5) | | |
| Tense and responsive | 22 (44.9) | 27 (55.1) | 22.87 | <0.001 |
| Very agitated and irritable, had an outburst | 9 (29) | 22 (71) | | |
| Calm and responsive | 42 (60.9) | 27 (39.1) | | |
| Calm and unresponsive | 3 (30) | 7 (70) | | |
| *Did the child have his or her blood drawn previously?* | | | | |
| No | 27 (52.9) | 24 (47.1) | 3.55 | 0.059 |
| Yes | 56 (37.8) | 92 (62.2) | | |
| *If the answer to the previous question was yes, how did he or she react? (No.: 148)* | | | | |
| Tense and unresponsive | 6 (26.1) | 17 (73.9) | | |
| Tense and responsive | 16 (51.6) | 15 (48.4) | | |
| Very agitated and irritable, had an outburst | 20.9 (31.7) | 43 (68.3) | 9.95 | 0.077 |
| Calm and responsive | 12 (50) | 12 (50) | | |
| Calm and unresponsive | 2 (28.6) | 5 (71.4) | | |

**Table 5.** Relationship between intellectual ability and anxious reaction.

| Variable | Intellectual Ability | | $\chi^2$ | *p* Value |
| --- | --- | --- | --- | --- |
| | Able | Disable | | |
| *How do the effects of anxiety appear in the child? (You may choose more than one answer. Check all that apply.)* | | | | |
| Repetitive behaviors | 38 (425.2) | 46 (54.8) | 1.23 | 0.266 |
| Social withdrawal | 34 (42) | 47 (58) | 0.09 | 0.762 |
| Anger, aggression, or violence toward oneself or others | 50 (39.4) | 77 (60.6) | 0.25 | 0.611 |
| Rigidity of thinking | 11 (39.3) | 17 (60.7) | 0.02 | 0.869 |
| Selective mutism | 11 (37.9) | 8 (62.1) | 0.1 | 0.742 |
| *How did the child react previously when visiting the doctor?* | | | | |
| Tense and unresponsive | 27 (67.5) | 13 (32.5) | | |
| Tense and responsive | 16 (32.7) | 33 (67.3) | | |
| Very agitated and irritable, had an outburst | 16 (51.6) | 15 (48.4) | 25.05 | <0.001 |
| Calm and responsive | 16 (23.2) | 53 (76.8) | | |
| Calm and unresponsive | 6 (60) | 4 (40) | | |
| *Did the child have his or her blood drawn previously?* | | | | |
| No | 18 (35.3) | 33 (64.7) | 0.83 | 0.362 |
| Yes | 63 (42.6) | 85 (57.4) | | |
| *If the answer to the previous question was yes, how did he or she react? (No.: 148)* | | | | |
| Tense and unresponsive | 9 (39.1) | 14 (60.9) | | |
| Tense and responsive | 8 (25.8) | 23 (74.2) | | |
| Very agitated and irritable, had an outburst | 36 (57.1) | 27 (42.9) | 13.78 | 0.017 |
| Calm and responsive | 6 (25) | 18 (75) | | |
| Calm and unresponsive | 4 (57.1) | 3 (42.9) | | |

## 4. Discussion

Children are vulnerable and often require customized care and special attention to meet their medical needs. This is increasingly true for children with autism. This study summarized the most frequently reported healthcare-related stressors faced by children with ASD and their families. The hospital experiences of patients with ASD are mostly in outpatient clinics, highlighting the level of consideration required for clinic visits. The average waiting time was two hours, which was found to be a catalyst for anxiety-related outbursts in this study and previous studies [19]. The most frequently reported anxiety inductions were phlebotomies and the measurement of vital signs. In keeping with international findings, the caregivers mentioned hospital environment-specific factors that made their children more anxious, including long waiting hours, sensory overload, and poor waiting room structure [19].

The gender distribution in this study was 3:1 male to female, which is in line with the prevalence of ASD across the international literature [20]. While global studies have delineated differences in the presentation of ASD across the genders, such as females having better socialization skills and fewer repetitive behaviors [20], no such findings were established regarding the presentation of anxiety in the healthcare setting in this study. We found no significant differences in anxiety among the age groups during doctor visits and increased anxiety in response to phlebotomies in school-aged children. However, conflicting data exist in the literature regarding the relationship between the age of the children and anxiety [21,22]. A study on generalized and social anxiety by Varela et al. (2020) [22] found that older children (12–18 years old) expressed more symptoms of overall and social anxiety than younger children (6–11 years old), while younger children expressed more generalized anxiety symptoms. Their findings proved consistent with the theory of differential expression of anxiety symptoms across ages. Davis et al. (2011) [21] used verified anxiety-scoring scales to explore the prevalence of anxiety symptoms in people diagnosed with autism across a lifespan. They concluded that anxiety symptoms are curvilinear in that they tend to increase from toddlerhood to childhood, decrease from childhood to young adulthood, and increase again from young adulthood to older adulthood. These studies draw conclusions on the prevalence of anxiety symptoms across age groups, and to the best of our knowledge, no previous study has discussed the prevalence of healthcare-related anxiety across different age groups. The contrast between our findings and the past literature on anxiety in children with autism may be due to the fact that our findings apply distinctly to the factors surrounding the doctor's office setting and phlebotomies, such as the change in environment and increased sensory stimulation. They are not intended to be conclusive of the prevalence of general anxiety symptoms between age groups in this population. Further studies exploring this relationship are recommended.

Anxiety in both preschool and school-aged individuals is lower in those with a higher IQ or less severe forms of autism [23]. A study on the psychological preparation of children with developmental disorders agreed that children with intellectual disabilities have an increased tendency to experience fear or agitation in hospitals associated with less favorable healthcare experiences [9]. In our study, there was a significant difference between the groups in reactions to doctor visits and reactions to phlebotomies. However, further inquiry into these differences can better assist in the modification of clinic visits and phlebotomies to make them more accommodating to this subgroup.

The participants who were nonverbal reported a higher occurrence of anger, aggression, and violence than their verbal counterparts during anxious situations, as well as an increase in outbursts. When studying the prevalence of anxiety in children with autism and minimal to no verbality, Tarver et al. (2021) [24] found that the parents of this group have unique challenges in identifying the triggers and manifestations of their children's anxiety. This can cause the child to rely on behavioral manifestations (i.e., aggression) to express feelings of anxiety. Their recommendations to manage anxiety for upcoming appointments are geared toward guardians, highlighting the importance of preparation by using visual imagery and scheduling of the appointment details, verbal repetition of the upcoming

appointment date, and understanding the child's sensory needs to provide them with the adequate tools needed to ease their transition [24]. To manage anxiety while in the hospital setting, they advise distraction with the use of the child's favorite toys, allowing them time and space to calm down, and offering reassurance in a way where the guardian is aware to be soothing for the child [24].

To address the challenges faced by children with autism and their caregivers in healthcare settings, Pratt et al. (2012) [25] recommended adequate preadmission planning and follow-through by a designated member of staff and the caregiver, as well as adequate knowledge of the child's individual communication needs. The caregivers in our study identified different solutions that they believed would help their children cope better with the anxiety of being in the hospital. These included access to waiting rooms equipped with toys and electronics, a television screen, and room for gross motor activity. Patients may further benefit from the use of ear-protective devices, such as noise-cancellation headphones, to decrease sensory overload [26]. Their responses were similar to those in a cross-sectional study by Mukhamedshina et al. [12] where 50% of the participants believed that providing playgrounds or specific rooms that cater to their child's needs for a reduced sensory load would result in a more effective healthcare examination.

This study faced a number of limitations, such as insufficient data from participants outside Jeddah. Additionally, we provided the total number of responses without recording the methods used to obtain the data (i.e., phone calls or independently via text messages). Categorizing responses based on the method of response may highlight potential biases. The findings regarding guardian satisfaction with doctors and nurses' accommodation skills only serve as a general viewpoint and lack insight into their coping abilities in specific situations (i.e., distress during venipuncture). This is an area for future qualitative study. Further studies are required to verify our results with a multi-center design, possibly taking into account the influence of additional invasive procedures to increase accuracy of problem identification, as well as studies implementing our recommendations to test how the overall healthcare experience would improve. The aim, ultimately, is that the findings of this study can be informative to hospital administrations in tertiary centers in the Kingdom, allowing them to implement institutional changes resulting in hospitals that are "child-friendly" [15] for those with disabilities such as autism. Studies further exploring the relationship between age group and health care anxiety as well as the differences in anxiety across genders may further enhance the healthcare system's understanding of the struggles faced by this population and in turn enhance its ability to better serve them.

## 5. Conclusions

While most patients with ASD face some hardships during healthcare visits, patients who are nonverbal and those with intellectual disabilities may have a higher tendency toward anxiety-induced outbursts in healthcare settings. Long hours in hospital waiting rooms that are not well accommodated for the sensitivities of patients with autism and lack any kind of entertainment are major contributors to child anxiety in healthcare settings. This may be ameliorated by the incorporation of toys, electronic devices, and spaces for gross motor activities in waiting rooms, as well as the personal use of ear-protective devices. A line of communication between the doctor and the caregiver pre-visit allows for tailoring the consultation to the child's needs and should be considered in healthcare centers. The intent of these findings is to help establish evidence-based practice guidelines related to the care of patients with ASD at the hospital staff and institutional levels to create hospital environments that are "child-friendly" to children with autism spectrum disorder.

**Author Contributions:** Conceptualization, B.A.-J., S.A., H.A., R.A., S.S., A.A, R.B. and H.B.; methodology, B.A.-J., S.A., S.S., A.A. and R.B.; software, S.A. and R.A.; validation, B.A.-J.; formal analysis, S.A., R.A. and S.S.; data curation, B.A.-J., S.A., H.A., R.A., S.S., A.A., R.B. and H.B.; writing—original draft preparation, S.A., H.A., R.A., S.S., A.A., R.B. and H.B.; writing—review and editing, B.A.-J., S.A., H.A., R.A. and H.B.; visualization, S.A., H.A., R.A. and S.S.; supervision, B.A.-J.; project administration, B.A.-J. All authors have read and agreed to the published version of the manuscript.

**Funding:** This research received no external funding.

**Institutional Review Board Statement:** This study was conducted in accordance with the Declaration of Helsinki and approved by the Ethics Committee of King Abdulaziz University (Approval No.: 157-21; date of approval: 23 March 2021).

**Informed Consent Statement:** Informed consent was obtained from all participants involved in this study.

**Data Availability Statement:** The data presented in this study are included in the tables. Additional data are available on request from the corresponding author. The data are not publicly available due to patient privacy.

**Acknowledgments:** The authors acknowledge the contributions of the autism rehabilitation centers, Tawahud Steps, Alamal Center, and parents in autism support groups in the questionnaire distribution process.

**Conflicts of Interest:** The authors declare no conflict of interest.

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
