# Peer review of "Healthcare Experience of Pediatric Patients with Autism Spectrum Disorders in Saudi Arabia: A Cross-Sectional Study"

_pediatrrep, doi:10.3390/pediatric15030042_

Round 1

Reviewer 1 Report

Title: Healthcare Experience of Pediatric Patients with Autism Spectrum Disorders in Saudi Arabia: A Cross-Sectional Study

This cross-sectional study aimed to gain insights into the healthcare experience of pediatric patients with Autism Spectrum Disorders (ASD) in Saudi Arabia by utilizing questionnaires distributed to caretakers and guardians. While the study provides valuable insights, minor revisions and a greater emphasis on clinical implications would enhance its impact.

1.       For the first sentence in the introduction section, "Autism Spectrum Disorder (ASD) is a neurodevelopmental disorder that significantly affects communication and social interactions..." (Lines 37-39), a citation is required.

2.       Were any measures taken by the authors to account for multiple comparisons? Given the numerous comparisons involved in the study, adjusting the level of significance would decrease the likelihood of committing a type I error.

3.       It would be beneficial to provide information on the number of parents and guardians contacted and the percentage of completed questionnaires. This data would allow us to understand the extent of non-response and refusal rates among parents.

4.       Please specify the number of parents who directly completed the questionnaire and the number of those who completed it through a phone call. Additionally, it would be insightful to explore whether there were any significant differences in responses between the two methods. For example, participants might have provided more positive answers during phone interviews compared to direct completion.

5.       In the discussion, the authors mentioned that Varela et al. (2020) proposed different findings from the current manuscript. It would be helpful to list the reasons the authors believe are responsible for the inconsistency in results.

6.       Based on the findings, please suggest clinical implications for both healthcare clinics and parents. These implications should focus on practical recommendations that can be derived from the study's outcomes.

7.       Please provide a comprehensive list of limitations and suggest potential directions for future research that can build upon this study's findings.

Author Response

[Comment 1] For the first sentence in the introduction section, "Autism Spectrum Disorder (ASD) is a neurodevelopmental disorder that significantly affects communication and social interactions..." (Lines 37-39), a citation is required.

Response: Thank you for taking the time to read and review our manuscript. Manuscript changes for this point can be found in:

  • Introduction (page 1, line 44)
  • References (page 11, line 568)

[Comment 2] Were any measures taken by the authors to account for multiple comparisons? Given the numerous comparisons involved in the study, adjusting the level of significance would decrease the likelihood of committing a type I error.

Response: Thank you for highlighting this point. After reviewing the indications for the Bonferroni correction, we believe that the statistical significance value should not be decreased beyond < 0.05 for any variables as they were compared individually with the use of one central hypothesis that children with ASD experience anxiety in the healthcare setting. For example, we compared age group with reaction to doctors’ visits, intellectual ability with presentations of anxiety, gender with method of communication. We did not combine multiple levels of comparison within any data in the analysis, so the p-value is unaffected. 

  • Reference: Sedgwick, Philip. (2012). Multiple significance tests: the Bonferroni correction. BMJ (online). 344. e509-e509. DOI: 1136/bmj.e509.

[Comment 3]: It would be beneficial to provide information on the number of parents and guardians contacted and the percentage of completed questionnaires. This data would allow us to understand the extent of non-response and refusal rates among parents.

Response: Thank you for this suggestion. We understand the relevance of the total number of people contacted in order to gain further insight from the refusal rate, but unfortunately we are unable to retrospectively access this number accurately as the questionnaire was only accessible to those who agreed to informed consent. We can provide the refusal rate of guardians who were contacted from the hospital database as well as the number of responses we did receive prior to exclusion of those who were above 18 years, however we have no record of those who refused from autism support groups or rehabilitation centers at this time, so the overall number would be inaccurate. The number of completed questionnaires was 199 responses, as can be found in: 

  • Results, Subsection: 3.1. Respondent characteristics (page 3, line 184)

[Comment 4] Please specify the number of parents who directly completed the questionnaire and the number of those who completed it through a phone call. Additionally, it would be insightful to explore whether there were any significant differences in responses between the two methods. For example, participants might have provided more positive answers during phone interviews compared to direct completion.

Response: Thank you for this note. We unfortunately did not include a question in the questionnaire to specify whether participants answered via phone call, in-person, or independently, so it is not possible for us to include this data into the paper. We recognize that this information can provide insight as to whether their responses were influenced - this is a limitation in the study. Manuscript changes for this comment can be found in:

  • Discussion (page 10, line 505)

[Comment 5] In the discussion, the authors mentioned that Varela et al. (2020) proposed different findings from the current manuscript. It would be helpful to list the reasons the authors believe are responsible for the inconsistency in results.

Response: Thank you for your suggestion. We adjusted the discussion to include the findings of both Varela et al. (2020) and Davis et al. (2011) to highlight some of what the literature has found regarding the difference in anxiety across age groups in children with autism. We further explained the lack of literature exploring healthcare-related anxiety across age groups makes it difficult to conclude whether our results are, at present, reproducible. The difference between our findings and past literature may simply be due to the difference in scope of anxiety explored; our results are not intended to make any conclusions about anxiety disorders as a whole in this population, rather the prevalence of healthcare-related anxiety. We further added a statement recommending further study on the topic to assess for validity. These added clarifications can be found in:

  • Discussion (page 9, lines 332, 340, and 347; page 10, line 517)

[Comment 6]  Based on the findings, please suggest clinical implications for both healthcare clinics and parents. These implications should focus on practical recommendations that can be derived from the study's outcomes.

Response: Thank you for your comment. We have mentioned top factors that increase children’s anxiety  being long waiting hours, overcrowdedness, and sensory overload. To tackle those we suggested to healthcare clinics the incorporation of waiting areas equipped with toys or electronics, decrease waiting hours and establishment of a line of communication between the doctor and guardian to tailor any specific needs. For parents, noise cancellation headphones can help decrease sensory overload. We further added the goal of implementing institutional changes by targeting hospital administrations, following further research, to develop the tertiary centers in the Kingdom to become more friendly for children with disabilities such as autism. The latter points are added to the manuscript in: 

  • Discussion (page 9, line 368; page 10, line 497 and line 514)
  • Conclusion (page 11, line 540-546)

[Comment 7]   Please provide a comprehensive list of limitations and suggest potential directions for future research that can build upon this study's findings.

Response: Thank you for the reminder, we have provided a list of limitations as well as added new limitations after reading reviewers’ points. Potential directions for future research are also added. Changes can be found in:

  • Discussion (page 10, line 504)

Reviewer 2 Report

in the document

Author Response

A. Materials and methods

[Comment A.1] The survey was then carried out either electronically or by telephone interview. It is necessary to explain the reasons for this different methodology. Does this have an impact on the nature of the responses?

Response: Thank you for taking the time to read and review our manuscript. We unfortunately did not include a question in the questionnaire to specify whether participants answered via phone call or independently, so it is not possible for us to include this data into the paper. We recognize that this information could have provided insight as to whether their responses were influenced by lack of clarity or the bias of speaking directly to healthcare worker - this is a limitation in the study. Manuscript changes for this comment can be found in:

  • Materials and Methods (page 2, line 83)
  • Discussion (page 10, line 505)

[Comment A.2] The questionnaire used is one created for this work. Did the construction of this questionnaire follow the rules of methodological elaboration that make it possible to verify sensitivity, reliability, and reproducibility?

Response: Thank you for your comment. The questionnaire used was validated by two developmental pediatrics consultants for content validity, and 10 parents for face validity. Manuscript changes for this comment can be found in:

  • Materials and methods (page 3, line 161)

[Comment A.3] When processing data from open-ended questions: how are these responses processed?

Response: Thank you for pointing this out. Content analysis was used. The open-ended question was “Does the child have skills to cope with anxiety? If your answer to the previous question was yes, what are these skills?” We viewed all the responses and created categories  of recurring themes and tallied the results accordingly. An addition was made to the manuscript for clarification, found in:

  • Materials and methods (page 3, line 178)

B. Results

[Comment B.1] This section is very descriptive. It needs to be revised to better highlight the results.

Response: Thank you for your comments. Further details were added to the results section.

  • Results section (page 5, line 237-245)

[Comment B.2] Paragraph 3.1: Are there differences in responses when it's moms alone who answer versus dad + mom or dad alone? Perceptions vary considerably, so this may be of interest.

Response: We appreciate this comment. To clarify, each child had either guardian respond to the questionnaire. No child had questionnaires completed by both guardians. As such, this is not a point of comparison in our study. We believe comparing the data set based on which guardian answered the survey will not yield an accurate point of comparison due to the vast differences in presentation between each child with ASD and their individual parents that cannot be accounted for through the information collected.

[Comment B.3] In the tables, the variables are presented for the total sample, but it is necessary to disaggregate by age group, as anxiety levels and stress-related behaviors vary considerably with age.

Response: Thank you for this note, we appropriately removed the sentence summary used to explain the differences in anxiety across age groups and replaced it with an additional table summarizing the our findings by age group. This can be found in the manuscript in:

  • Results, Subsection: 3.3. Characteristics of the healthcare experiences (page 5, line 255)
  • Results, Table 3 (page 5, line 277)

[Comment B.4] Similarly, the study shows a 4:1 ratio of boys to girls, reflecting the prevalence of ASD. However, the authors make no mention of this data. However, the literature reports a greater severity of ASD in girls. Consequently, it would be highly relevant to analyze this gender effect on the variables measured and on the severity of the clinical signs reported:

- For example: do girls, whatever their age, have more (or less?) manifestations of anxiety, aggressive behavior... during care?

To respond to these analyses, it is imperative that the authors revisit their statistical plan to enhance their results.

Response: We appreciate this comment. Analysis was done to compare the reactions to doctors visits and phlebotomy between genders and the results were nonsignificant (p=0.113 and p=0.618, respectively). These points have been added to the results of the manuscript as well as the discussion. To note, conflicting evidence on the gender disparaties in children with autism exists in the literature. Found in:

  • Materials and methods (page 3, line 165)
  • Results, Subsection: 3.3. Characteristics of the healthcare experiences (page 5, line 251)
  • Discussion (page 9, line 328)

[Comment B.5] The effect of the verbal/non-verbal variable is too inadequate. Moreover, is it mainly girls who make up the non-verbal group?

Response: Thank you for your comment. We included this analysis to the manuscript, as well as an addition to the discussion on the association between behaviour and verbal/non-verbal communication. This can be found in:

  • Results, Subsection: 3.3 Characteristics of the healthcare experience (page 5, line 258)
  • Discussion (page 9, line 366 and 368)

C. Discussion

[Comment C.1] This section is far too light and superfluous.
The authors discuss the effect of age, but the results are not presented in this sense.

Response: Additions have been made to the discussion section according to the previous comments. In addition, a table was added to the manuscript illustrating the results by age group more clearly. This can be found in:

  • Results, Table 3 (page 5, line 277)
  • Discussion (page 8)

[Comment C.2] Similarly, there is a focus on one case of phlebotomy, but no doubt other medical procedures were invasive.

Response: Thank you for the reminder, this note has been added to the limitations and areas of further research, which can be found in:

  • Discussion (page 10, line 511-512)

D. Additional

[Comment D.1] It would be very interesting to detail the types of care reported by parents as being adapted to the characteristics of a child with ASD. Recommendations would also be an added value:

- what not to do, as several parents have reported very deleterious effects on their child's behavior when he or she was in care,

- or, on the contrary, what was helpful in understanding the medical environment, because the care team had been attentive to this or that detail. 

These recommendations could therefore be very useful to healthcare professionals, but also to educators and families for everyday care gestures. 

Response:  Thank you for these suggestions. These are very valuable points and no doubt essential to create a smoother healthcare experience. However considering the focus of the project is on the healthcare experience itself, we believe adding these points to the questionnaire would add more variables than can be controlled in this paper. Recommendations for what guardians may benefit from doing have been added from the literature review to the discussion. Further exploration of what was helpful for understanding the healthcare experience and what guardians believe was helpful or not helpful by healthcare staff is an area of future qualitative study. Further, the coping skills mentioned in the results serve as additional strategies that can be implemented in the healthcare setting. These can be found in:

  • Results, Table 2 (page 4, line 224)
  • Discussion (page 9, line 368; page 10, line 497 and line 508)
  • Conclusions (page 10, line 526; page 11, line 540-543)

Round 2

Reviewer 2 Report

A precision must be added. Authors wrote:  The questionnaire used was validated by two developmental pediatrics consultants for content validity, and 10 parents for face validity. This methodological approach is not a validation method.

Authors must be very precise. 

This questionnaire has not been validated by the methodological and metrological steps. Authors must be honest and simple. It's a 'local' questionnaire. 

Author Response

Thank you for the comment. To clarify, we used a previously validated questionnaire by Muskat et al as a guide. We then performed extensive literature review to develop a questionnaire that targets the factors deemed relevant to test our hypothesis. It was reviewed by 2 developmental pediatrics consultants for content validity and by 10 local parents for face validity. We would like to emphasize our adherence to the rules and regulations of research development; every single step was reviewed and approved by the Research Ethics Committee of our institution (Approval No. 157-21), including questionnaire formulation and validation in both languages. Manuscript clarifications can be found in:

  • Materials and methods (page 3, lines 160-163)